# Glyphosate and AMPA in Human Urine of HBM4EU-Aligned Studies: Part B Adults

**DOI:** 10.3390/toxics10100552

**Published:** 2022-09-21

**Authors:** Jurgen Buekers, Sylvie Remy, Jos Bessems, Eva Govarts, Loïc Rambaud, Margaux Riou, Thorhallur I Halldorsson, Kristin Ólafsdóttir, Nicole Probst-Hensch, Priska Ammann, Till Weber, Marike Kolossa-Gehring, Marta Esteban-López, Argelia Castaño, Helle Raun Andersen, Greet Schoeters

**Affiliations:** 1Unit Health, VITO, Flemish Institute for Technological Research, 2400 Mol, Belgium; 2Department of Environmental and Occupational Health, Santé Publique France, 94415 Saint-Maurice, France; 3Faculty of Food Science and Nutrition, University of Iceland, 102 Reykjavik, Iceland; 4Faculty of Medicine, University of Iceland, 102 Reykjavik, Iceland; 5Swiss Tropical and Public Health institute, 4123 Allschwil, Switzerland; 6Department of Public Health, University of Basel, 4056 Basel, Switzerland; 7German Environment Agency (UBA), 06844 Dessau-Roßlau, Germany; 8Instituto de Salud Carlos III, National Centre for Environmental Health, 28220 Madrid, Spain; 9Department of Public Health, University of Southern Denmark, 5000 Odense, Denmark; 10Department of Biomedical Sciences, University of Antwerp, 2610 Antwerp, Belgium

**Keywords:** HBM4EU, glyphosate, AMPA, exposure, HBM, adults

## Abstract

Within HBM4EU, human biomonitoring (HBM) studies measuring glyphosate (Gly) and aminomethylphosphonic acid (AMPA) in urine samples from the general adult population were aligned and quality-controlled/assured. Data from four studies (ESB Germany (2015–2020); Swiss HBM4EU study (2020); DIET-HBM Iceland (2019–2020); ESTEBAN France (2014–2016)) were included representing Northern and Western Europe. Overall, median values were below the reported quantification limits (LOQs) (0.05–0.1 µg/L). The 95th percentiles (P95) ranged between 0.24 and 0.37 µg/L urine for Gly and between 0.21 and 0.38 µg/L for AMPA. Lower values were observed in adults compared to children. Indications exist for autonomous sources of AMPA in the environment. As for children, reversed dosimetry calculations based on HBM data in adults did not lead to exceedances of the ADI (proposed acceptable daily intake of EFSA for Gly 0.1 mg/kg bw/day based on histopathological findings in the salivary gland of rats) indicating no human health risks in the studied populations at the moment. However, the controversy on carcinogenicity, potential endocrine effects and the absence of a group ADI for Gly and AMPA induce uncertainty to the risk assessment. Exposure determinant analysis showed few significant associations. More data on specific subgroups, such as those occupationally exposed or living close to agricultural fields or with certain consumption patterns (vegetarian, vegan, organic food, high cereal consumer), are needed to evaluate major exposure sources.

## 1. Introduction

Glyphosate (Gly; C3H8NO5P) was initially developed as a chelating agent to remove mineral deposits in water pipes [1]. Later, it was patented as a herbicide for agricultural use. Now, it is the most widely used plant protection product worldwide [2]. In Argentina for example, Gly use has increased over the years, especially in the Pampas region, typically for soy production. In the EU, Gly is currently approved until December 2022 [3]. Already in 1993, Gly was detected in human urine in the US [4]. Overall, human exposure data in the EU are limited [5] and health effects, especially cancer, remain unclear up to this point. IARC has classified Gly as probably carcinogenic (Group 2A) [6], while the EU regulatory agencies consider its aquatic toxicity as the basis for regulation [7]. More human exposure data and epidemiological studies analysing associations with health effects are warranted. HBM4EU (www.hbm4eu.eu; accessed on 1 August 2022) addressed the human exposure side of the problem, and human biomonitoring (HBM) data on Gly and AMPA (main metabolite of Gly and main degradation product of glyphosate in the environment) were analysed in adults and children of various EU countries. This manuscript contains the results for adults. The results for children are described in detail in Buekers et al. (2022) [8].

## 2. Methods

### 2.1. Data

Data on Gly and AMPA in urine of adults were obtained in HBM4EU-aligned studies from Germany (UBA ESB; 2015–2020), Switzerland (Swiss TPH HBM4EU study; 2020), France (ESTEBAN; 2014–2016) and Iceland (DIET-HBM; 2019–2020). The study designs are described in Gilles et al. (2021) [9] and characteristics of study participants and ethics can be found in Gilles et al. (2022) [10]. The Swiss HBM4EU study, DIET-HBM and ESTEBAN are nationally representative campaigns.

Samples were analysed by harmonised methods in sometimes different laboratories. Biomarker data were quality-assured/controlled following a strict schedule [11]. All data were included for analysis of exposure determinants and risk assessment. Values below the LOQ (limit of quantification) were not imputed for Gly or AMPA because quantification rates were low (below 50% in most studies).

Samples with creatinine concentration ≥5 mg/dL and a specific gravity of 1.001 to 1.020 were defined as acceptable. Urine of healthy persons would be unlikely to be excluded using these criteria [12]. These criteria were used for all data.

In the German study, Gly and AMPA were measured in 24 h urine, whereas in the other studies, spot urine or first morning urine samples were used. Spot urine and morning urine give a glimpse of exposure to a substance and might not be representative for 24 h exposure when being calculated to the total excreted amount for a day. This is especially a limitation for relatively short half-life chemicals, such as Gly (half-life 5–10 h, [13]). Instead of recalculating all data to a 24 h scale, we decided to use the measured concentrations as such (independent of the type of urine sample) for addressing associated exposure determinants.

Potential determinants of exposure variability were obtained through questionnaires from the specific studies. Questionnaires collected within the new and ongoing HBM campaigns under HBM4EU were harmonized. As some HBM campaigns were ongoing, variables were harmonized post factum. As a consequence, not all studies provided information on the same variables. For upcoming HBM studies, basic and substance-specific questionnaires are made available and can be used [14].

### 2.2. Identifying Determinants of Exposure

Statistical analyses were performed in SPSS Statistics 28. In a first step, the individual studies were analysed separately. For assessing the determinants of variability of Gly and AMPA concentrations, both variables were dichotomized (value 0 for values below LOQ and value 1 for values above according to the LOQs in the individual studies). A logistic regression model was applied to analyse exposure determinants. Single variables (or determinants) which might explain differences in Gly or AMPA concentrations were studied with matrix (spot, morning or 24 h urine), BMI and creatinine forced into the model. BMI was forced into the model because of the possible association with creatinine. Determinants considered were individual characteristics, dietary preferences, exposure relevant behaviour (e.g., use of pesticides) and sociodemographic information (see Appendix A for detailed information). Secondly, a logistic multiple regression model was built including all variables with *p*-value ≤0.2 in the former analysis for each study separately. By backward selection, starting with excluding the variable with highest *p*, only those variables with *p* ≤ 0.05 were kept in the analysis. Thirdly, in a final logistic regression model, data from all studies were combined and analysed as a single dataset. A value of 0.1 µg/L was applied as the LOQ limit (threshold) for all studies. Country was included in the model as a covariate. Determinants of variability were assessed one at a time (see Appendix A). No logistic multiple regression with backward selection was performed for all studies combined as not all determinants were available for all studies (see Appendix A).

### 2.3. Risk Assessment

IARC classified Gly as probably carcinogenic (group 2A) [6] while ECHA classified Gly again as a non-carcinogen [15]. This controversy remains debated. No HBM guidance value for exposure to Gly exists yet. Based on reversed dosimetry, external exposure values were estimated for Gly [16]. The predicted daily intake (PDI) was compared with the acceptable daily intake (ADI). The main assumptions made are described in Buekers et al. (2022) [8]. Briefly, for calculation of the PDI as percentage of the ADI, the urine concentration of Gly was multiplied by the volume of urine, divided by the body weight multiplied by urinary excretion fraction (F_UE_) of Gly and by the proposed ADI of EFSA (Equation (1)). The F_UE_ is the urinary excretion fraction. It is the ratio of the mass of glyphosate excreted in urine over the mass of glyphosate ingested assuming a constant mass balance. This results in the predicted daily intake of Gly expressed as percentage of the ADI.
(1)$$\%ADI=\frac{Gly_{conc\;}\times Vol_{urine}}{bw\times F_{UE}\times ADI}=\frac{PDI}{ADI}$$
where Gly_conc_ is the concentration of glyphosate measured in urine; Vol_urine_ is standardized as 2 L/day for adults; bw is bodyweight which is standardized at 75 kg; the F_UE_ is set at 0.57% [17]; and ADI is the acceptable daily intake allowance for Gly (proposed by EFSA to be reduced to 0.1 mg/kg bw/day; Point of Departure, PoD = NOAEL of 10 mg/kg/day based on histopathological findings in the salivary gland in a 2-year rat study, to which a standard assessment factor of 100 was applied [18]). The current ADI of EFSA is 0.5 mg/kg bw/day, but it is proposed by EFSA to reduce it to 0.1 mg/kg bw/day [18]. Using urinary P95 values for Gly_conc_ (reasonable worst case), we calculated PDI values and compared these to the ADI.

## 3. Results and Discussion

### 3.1. Exposure

An overview of Gly and AMPA measured in the HBM4EU-aligned studies is presented in Table 1 and Table 2. The results are discussed and compared with published studies from Europe found in the international literature (Table 3).

HBM4EU data showed that the exposure is widespread in the EU. In general, median concentrations below LOQ were observed in the HBM4EU studies for Gly and AMPA (Table 1 and Table 2). At the higher end of the exposure distribution, the highest concentrations (µg/L) were observed for Gly in Iceland (P95 0.37 µg/L) and for AMPA in France (P95 0.42 µg/L) (Table 1 and Table 2). When data are compared to existing data (Table 3), P95 values for Gly varied between 0.24 and 1.25 µg/L and for AMPA between 0.21 and 1.54 µg/L. The greatest P95 values for Gly (1.25 µg/L) and AMPA (1.54 µg/L) were from the study of Conrad et al. (2017) for the year 2013 (years sampled: 2001 to 2015). When a comparison is drawn with data in children [8] of the HBM4EU aligned studies, the P95 were in the same order of magnitude for children and adults but higher in children (results of combined data regardless of result QA/QC: Children: Gly 0.51 µg/g crt or 0.48 µg/L and AMPA 0.45 µg/g crt or 0.47 µg/L; Adults: Gly 0.30 µg/g crt or 0.29 µg/L and AMPA 0.30 µg/g crt or 0.33 µg/L (Table 1 and Table 2)). In Figure 1, P50 and P95 values are presented for children and adults of the different studies. For France, it was shown in the study of ESTEBAN that the P95 for Gly was equal to 0.35 µg/g crt (95%CI: 0.15 to 0.75) in adults and 0.84 µg/g crt (95% CI: 0.53 to 1.51) in children. For AMPA, this was 0.38 µg/g crt (95%CI: 0.29 to 0.44) in adults and 0.59 µg/g crt (95%CI: 0.48 to 0.76) in children. Children have a higher ingestion of food and drink per kilogram bodyweight in relation to adults. They play outdoors and are probably more exposed to pesticides in the outdoor environment. The difference in P95 between children and adults shows that care should be taken for susceptible groups like children. Although the results represent the selected studies, they give an indication of the EU exposure. HBM4EU substantially contributed to providing exposure data within the EU, nevertheless, exposure data on Gly and AMPA in children, especially those living close to agricultural fields, remain limited even though they are a vulnerable population. Limited exposure data from the US showed arithmetic mean values between 0.28 and 0.61 µg/L in children [26,27,28] with even a geometric mean value in one study of 2.5 µg/L for children of non-farming households [29]. Concentrations were higher in children than in adults of the US, however, only limited data were available [28,29].

### 3.2. AMPA to Glyphosate Ratio (AMPA/Gly)

In the HBM4EU-aligned studies, the ratio at molar basis of AMPA vs. glyphosate was analysed for samples >LOQ in adults (*n* = 97) (Figure 2). The AMPA/Gly ratio for creatinine-corrected concentrations varied between 0.2 and 9.0, with an average value of 1.8. The ratio decreased with increasing Gly. When regressing AMPA (ln µg/g creatinine) on the *Y*-axis against Gly (ln µg/g creatinine) on the *X*-axis, the slope of the linear fit is smaller than 1 (*p* < 0.001). When the regression line is compared with results in children (Figure 3), it can be seen that these are similar. Slopes vary between 0.35 and 0.39 and the intercept between −3.9 and −3.8 (Figure 2 and Figure 3).

The intercept on the left shows that at low Gly concentrations, there is still AMPA present in urine. This observation indicates the existence of ‘autonomous’ origins of AMPA (independent of metabolism of GLY to AMPA in the monitored participants). As discussed in Buekers et al. (2022), this may be a consequence of AMPA being present in the environment from glyphosate or from other sources such as amino-polyphosphonates [30].

### 3.3. Risk Assessment (RA)

There is still a conflict of opinions between IARC and the EU agencies on the carcinogenicity of Gly, as the ECHA RAC has just concluded in May 2022 in its scientific opinion that it is not justifiable to classify GLY as carcinogenic. The assumption made regarding hazard identification might very well influence the hazard characterization due to differing sensitivities to different endpoints. The proposed ADI of 0.1 mg/kg bw/day is not exceeded. For the HBM4EU studies, the %ADI was at maximum 1.73%, starting from P95 exposure values (Table 4). In a farmworker population and a general population living close to agricultural fields, the predicted daily intake may be closer to the proposed ADI. A group ADI is proposed, seeing a similar toxicological profile for AMPA and Gly [31,32].

Associations between health effects and exposure to Gly have been discussed in multiple studies. There are controversies on the carcinogenicity and endocrine disruption [6,33,34] and recent findings show associations with reproductive effects [35,36,37,38]. Additionally, the gut microbiome may be influenced [39,40]. Glyphosate affects a metabolic pathway in plants (shikimate) that human cells do not have. However, gut microbes possess this pathway to synthesize tryptophan, tyrosine and phenylaniline necessary for building human proteins, including vitamin B and neurotransmitters. A conservative estimate from Leino and colleagues shows that 54% of species in the core human gut microbiome are sensitive to Gly [41]. Gut health may also have an influence on brain function via the gut-brain axis [42]. Overall, there is a need for new studies exploring the links between glyphosate and human health effect markers at environmentally relevant concentrations alone or in combination with other pollutants, searching for links with observed effects found in animal and in vitro studies (e.g., Vardakas et al., 2022) [43].

### 3.4. Exposure Determinants

Biomarker concentrations of urinary Gly and AMPA levels were higher in the HBM4EU-aligned studies for children compared to adults. Moreover, exposure determinants might differ between children and adults, given differences in dietary consumption patterns and behaviour.

Among the HBM4EU-aligned studies, the logistic multiple regression model—adjusted for sex, BMI, crt—in adults did show that for the German ESB study, a higher frequency of consumption of fruit and vegetables resulted in higher concentrations of internal Gly (Table 5). When sex was removed from the model, the frequency of fruit/vegetable consumption was not significant anymore (*p* > 0.05). Glyphosate concentrations were significantly higher in males than in females in UBA ESB (See Appendix A). For the Swiss study, the logistic multiple regression model did show lower Gly concentrations in employed persons compared to non-employed persons. The reason here is unknown, but there is a significant positive association between ISCED and occupational status. Differences in socioeconomic status, for which ISCED is an indicator, may have an influence on chemical exposure [44]. Finally, the study in Iceland did show higher concentrations in vegetarians, although the number of vegetarians was small (*n* = 5) and the confidence interval on the odds ratio (OR) was large (Table 5). All models remained significant when BMI was omitted from the model. No other significant associations (*p* < 0.05) were observed in the logistic multiple regression models with backward selection.

No significant associations with the consumption of cereals, a known source of Gly, was found (Appendix A). For the combined studies, no significant effect of fruit and vegetable consumption on internal Gly or AMPA concentrations was found. Overall, when data of the different studies were combined, logistic regression did not show clear and significant associations with exposure determinants.

Exposure determinants in adults have been reported previously. In a cross-sectional study, Soukup et al. (2020) reported a positive correlation between the consumption of pulses and urinary Gly levels, based on quantifiable amount of Gly in 8% (*n* = 25) of the samples [21]. In Europe, the highest Gly loads in food were reported for pulses and cereals, although only 2–3% of the food items contained quantifiable amounts of Gly between 2015 and 2017 [45,46,47,48]. In Denmark, Gly residues have been found in barley, wheat grain, wheat flour, oat grain, cornflakes, dried lentils and chickpeas, all below the current maximum residue level [24]. Conrad et al. (2017) found in their study of participants in the ESB (Environment Specimen Bank) that vegetarians/vegans (*n* = 10; low participation of males) did not have higher concentrations of Gly or AMPA, which was against the expectations [23]. In an older German and Danish study, significantly higher concentrations of Gly were observed in people consuming conventional food versus people who ate predominantly an organic diet [49]. Ruiz et al. (2021) did find that eggs and fruit were main predictors for Gly exposure in adults [19]. High frequency of physical exercise or a residence remote from agricultural activities were also associated with lower levels of Gly. The tendency with physical exercise was also found in the US study of John and Liu (2018), although not to a significant degree [50]. John and Liu (2018) did observe elevated levels of Gly in participants who had consumed tea in the past 24 h (US study) [50].

For children, associations were found with sex, age, BMI, distance to agricultural fields, sampling season, use of local food, consumption of nuts, wholegrain, educational level, parental pesticide use and presence of pets in the home [8,24,51,52]. These findings were not always consistent. There were also studies that did not find differences in Gly and AMPA exposure between population subgroups [48]. Hereby, it is observed that there are sometimes gaps in information about exposure determinants or that extra information is needed, e.g., detailed information on distance of living to agricultural fields, type of crops, application time and type of plant protection products, being vegetarian, frequency of organic food consumption, and this is required for a large subgroup of people to obtain enough statistical power to register the differences.

## 4. Conclusions

HBM4EU collected quality-controlled/assured HBM data on Gly and AMPA adult exposure in the EU. Overall, median values were below the reported LOQs in adults. The results are in line with what was found in published European studies. Children clearly experience higher exposure than adults. Data on the AMPA/Gly ratio support the existence of autonomous sources of AMPA. The discussion on the carcinogenic effects of Gly is still ongoing and risk assessment for non-carcinogenic effects comparing the ADI with the PDI did not show risks of Gly exposure in adults at the moment. Exposure determinants analysis showed some evidence that the consumption of fruit and vegetables and work status may play a role in the differences in observed Gly concentrations in the individual studies. However, a combination of data of different studies could not confirm this finding. More specific data on certain subgroups of persons living close to agricultural fields or being frequent consumers of certain food categories remain necessary, with an adequate number of people in the specific categories.

## Figures and Tables

**Figure 1 toxics-10-00552-f001:**
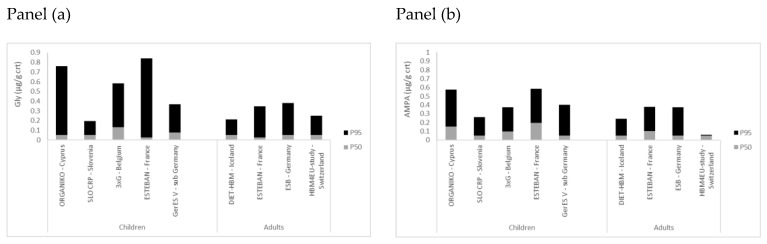
Glyphosate (Gly) and AMPA (µg/g crt) in adults and children of the aligned studies under HBM4EU. Panel (**a**) Gly and panel (**b**) AMPA. Data for children described in Buekers et al., 2022 [8]. Values below the LOQ were set at the LOQ/2 for presentation.

**Figure 2 toxics-10-00552-f002:**
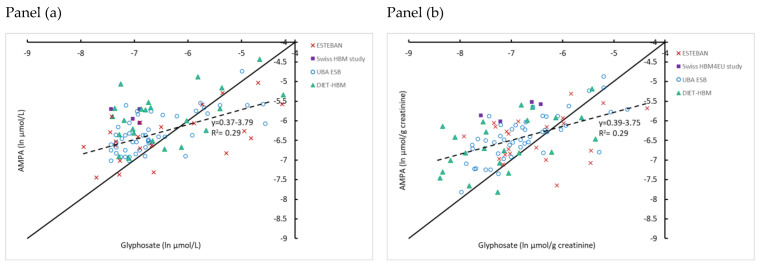
HBM4EU harmonized data (Germany (UBA ESB), Switzerland (Swiss HBM4EU study), France (ESTEBAN), Iceland (DIET-HBM)) for glyphosate and AMPA in adults. Data (>LOQ) are expressed on a natural logarithmic scale. The 1:1 line (full line) is indicated. Panel (**a**): µg/L scale and panel (**b**): µg/g creatinine scale. Values below LOQ are not included.

**Figure 3 toxics-10-00552-f003:**
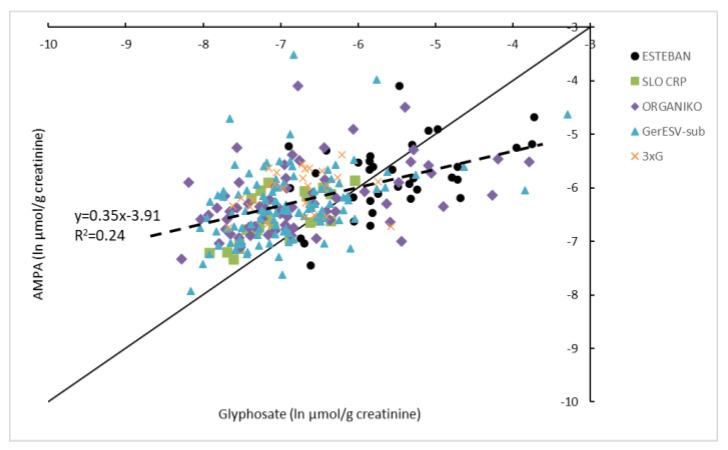
HBM4EU harmonized data (Belgium (3XG), Cyprus (ORGANIKO), Germany (GerES V-sub), Slovenia (SLO_CRP), France (ESTEBAN)) for glyphosate and AMPA in children. Data (>LOQ) are expressed on a natural logarithmic scale and corrected for creatinine. The 1:1 line (full line) is indicated. See also [8] for more information on the children’s studies.

**Table 1 toxics-10-00552-t001:** Urinary glyphosate (Gly) concentrations in adults from HBM4EU-aligned studies.

Study	Sampling Year	N	Age Range (Years)	Urine Sample	Method	LOD Gly (µg/L)	LOQ Gly (µg/L)	%<LOQ ^d^	Gly (µg/L)				Creatinine-Adjusted Gly (µg/g_crt_) ^e^				Creatinine Median (P5 and P95) in mg/dL
									P25	P50	P75	P95	P25	P50	P75	P95	
UBA ESB (Germany) ^a^	2015–2020	250	20–29	24 h	GC/MS-MS		0.1	70	<LOQ	<LOQ	0.11	0.29			0.10	0.38	70 (33–174)
Swiss HBM4EU study (Switzerland) ^b^	2020	299	20–39	Morning	LC/MS-MS		0.1	81	<LOQ	<LOQ	<LOQ	0.24				0.25	98 (31–250)
DIET-HBM (Iceland) ^b^	2019–2020	195	20–39	Spot	GC/MS-MS		0.1	71	<LOQ	<LOQ	0.12	0.37			0.05	0.21	127 (22–284)
ESTEBAN (France) ^a^	2014–2016	169	20–39	Morning	LC/MS-MS	0.02	0.05	83	<LOQ	<LOQ	<LOQ	0.24				0.34	102 (26–215)
Combined data ^c^												0.29				0.30	

^a^: Biomarker data generated before HBM4EU QA/QC program and comparability cannot be guaranteed (see Esteban López et al., 2021) [11]. ^b^: Biomarker data quality assured by HBM4EU QA/QC program. ^c^: results of all datasets combined regardless of result QA/QC program. ^d^: Each study was compared to its own LOQ. ^e^: Values <LOQ set to LOQ/2.

**Table 2 toxics-10-00552-t002:** Urinary AMPA concentrations in adults of the HBM4EU-aligned studies.

Study	LOD (µg/L)	LOQ (µg/L)	%<LOQ ^c^	AMPA (µg/L)				Creatinine-Adjusted AMPA (µg/g_crt_) ^d^			
				P25	P50	P75	P95	P25	P50	P75	P95
UBA ESB (DE) ^a^		0.1	66	<LOQ	<LOQ	0.13	0.34			0.13	0.37
Swiss HBM4EU study (CH) ^b^		0.1	96	<LOQ	<LOQ	<LOQ	<LOQ				
DIET-HBM (IS) ^b^		0.1	77	<LOQ	<LOQ	<LOQ	0.37				0.24
ESTEBAN (FR) ^a^	0.02	0.05	24	0.06	0.10	0.18	0.42	0.05	0.10	0.18	0.36
Combined data							0.33				0.30

^a^: Biomarker data generated before HBM4EU QA/QC program but deemed comparable (see Esteban López et al., 2021) [11]. ^b^: Biomarker data quality assured by HBM4EU QA/QC program. ^c^: Each study was compared to its own LOQ. ^d^: Values <LOQ set to LOQ/2.

**Table 3 toxics-10-00552-t003:** Comparison with published European exposure data on urinary glyphosate (Gly) and AMPA concentrations in EU adults from the general population.

Study	Country	Sampling Year	Population	Urine Sample	Method	Gly in µg/L			AMPA in µg/L		
						LOQ/LOQ	Average	P95	LOQ/LOQ	Average	P95
Ruiz et al., 2021 [19]	Spain	2016	97 breastfeeding mothers 25–45 y	Morning urine	LC/MS-MS	LOQ = 0.1	GM 0.12	0.62	LOQ = 0.1	GM 0.14	0.69
Faniband et al., 2021 [20]	Sweden	2017	197 adults 18–19 y	Spot	LC/MS-MS	LOD = 0.1	Median < LOD	0.24	LOD = 0.1	Median < LOD	0.25
Soukup et al., 2020 [21]	Germany	2012–2013	301 adults 18–80 y	24 h	LC/MS-MS	LOQ = 0.2	Median < LOQ	NA	LOQ = 0.2	Median < LOQ	NA
Connolly et al., 2018 [22] **	Ireland	2017	50 adults, 18–82 y	Spot	LC/MS-MS	LOQ = 0.5	Median < LOQ	NA	NA	NA	NA
Conrad et al., 2017 [23] *	Germany	2001–2015	399 adults, 20–29 y	24 h	GC/MS-MS	LOQ = 0.1	Median < LOQ to 0.11	0.12 to 1.25	LOQ = 0.1	Median < LOQ to 0.12	0.21 to 1.54
Knudsen et al., 2017 [24]	Denmark	2011	13 Mothers, 31–52 y	Spot	ELISA	LOD = 0.0751	AM 1.28	NA	NA	NA	NA
Hoppe et al., 2013 [25]	Europe	2013	182 adults	Spot	GC/MS-MS	LOQ = 0.15	Median < LOQ	0.92	LOQ = 0.15	Median < LOQ	0.64

AM: arithmetic mean; GM: geometric mean; P95: 95th percentile; NA: not available. *: The study of Conrad et al. (2017) [23] covers different samples years over the period 2001 to 2015. **: Further studies are currently evaluating whole-family exposures (www.nuigalway.ie/image; accessed on 1 August 2022).

**Table 4 toxics-10-00552-t004:** Risk assessment of glyphosate (Gly) in adults of the general population based on HBM data (detection of Gly and AMPA with mass spectrometry).

Ref.	P95 Concentration Glyphosate	PDI	PDI	PDI/ADI
	µg/L	µg/day	µg/kg bw/day	%
UBA ESB (DE)	0.29	=0.29 × 2/0.57% = 102	=102/75 = 1.36	=0.00136/0.1 = 1.36%
Swiss TPH (CH)	0.24	84	1.12	1.12%
DIET-HBM (IS)	0.37	130	1.73	1.73%
ESTEBAN (FR)	0.24	84	1.12	1.12%
Previous studies				
Ruiz et al., 2021 [19]	0.62	218	2.90	2.90%
Faniband et al., 2021 [20]	0.24	84	1.12	1.12%
Conrad et al., 2017 [23] *	1.25	439	5.85	5.85%
Hoppe et al., 2013 [25]	0.92	323	4.30	4.30%

PDI: predicted daily intake; ADI: acceptable daily intake. Assumed bodyweight set at 75 kg, urinary volume at 2 L/day and F_UE_ at 0.57%. ADI was set at 0.1 mg/kg bw/day. *: year 2013 selected.

**Table 5 toxics-10-00552-t005:** Logistic backward multiple regression glyphosate (Gly) and AMPA HBM4EU data.

	Study	Variable	OR = Exp(β)	95%CI	*p* Value
Gly	UBA ESB	Intercept	0.03		0.103
		Creatinine	1.02	(1.01;1.03)	<0.001
		BMI	0.99	(0.89;1.11)	0.877
		Sex	2.77	(1.42;5.41)	**0.003**
		Frequency consumption fruit/vegetables: sometimes and often *	Ref (*n* = 41)		**0.07**
		Frequency consumption fruit/vegetables: very often	1.33 (*n* = 62)	(0.48;3.70)	
		Frequency consumption fruit/vegetables: everyday	2.51 (*n* = 147)	(1.00;6.28)	
		Model			<0.001
	Swiss HBM4EU study	Intercept	1.08		
		Creatinine	1.01	(1.00;1.01)	<0.001
		BMI	0.91	(0.82;1.00)	0.05
		Occupational status (not employed)	Ref (*n* = 58)		
		Occupational status (employed)	0.39 (*n* = 239)	(0.20;0.77)	**0.007**
		Model			<0.001
AMPA	DIET-HBM	Intercept	0.12		0.096
		Creatinine	1.01	(1.01;1.02)	<0.001
		BMI	0.99	(0.92;1.08)	0.973
		Not vegetarian	Ref (*n* = 167)		
		Vegetarian	7.68 (*n* = 5)	(1.01;58.28)	**0.049**
		Model			<0.001

OR: odds ratio. Reference or Exp(B) = 1. BMI, creatinine (crt) and matrix (morning, spot, 24 h urine) were forced into the model. *: Redistribution categories: “sometimes (*n* = 12)“ and “often (*n* = 29)” were combined and compared to “very often (*n* = 62)” and “everyday (*n* = 147)”. Only multiple regression models with *p* of less than or close to 0.05 shown in table.

## Data Availability

The data supporting the results can be found at www.HBM4EU.eu and https://www.hbm4eu.eu/what-we-do/european-hbm-platform/eu-hbm-dashboard/ (accessed on 1 August 2022).

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
