# Peer review of "Glyphosate and AMPA in Human Urine of HBM4EU-Aligned Studies: Part B Adults"

_toxics, 2022, doi:10.3390/toxics10100552_

Round 1
Reviewer 1 Report (Previous Reviewer 1)
All the comments were adressed.
Reviewer 2 Report (Previous Reviewer 3)
-
This manuscript is a resubmission of an earlier submission. The following is a list of the peer review reports and author responses from that submission.
Round 1
Reviewer 1 Report
The manuscript presents a comprehensive analyses of the implications of the discovery of glyphosate and AMPA in the urine of Europeans.
This manuscript is well written and presents novel findings, in particular the relationship between glyphosate and AMPA indicating the presence of novel sources and the inverse dosimetry calculation providing some indications of health risk implications.
Have you explored whether the relationship between glyphosate and AMPA is non-linear using non-linear statistical models?
Could you summarise briefly if the participants from this study are representative of the general populations? Many studies are biased for highly educated and high income participants.
Since a large number of urine samples were free from detectable glyphosate, have you tried considering glyphosate detection as a categorical variable (detected or not detected) in order to associate with the other covariate and understand the sources of exposure?
Even if the project is European, I feel that it would be worth including the recent study performed in the UK by Mesnage and colleagues who measured glyphosate and AMPA in the urine of 124 people (Mesnage R et al. Impacts of dietary exposure to pesticides on faecal microbiome metabolism in adult twins. Environ Health. 2022 May 3;21(1):46. doi: 10.1186/s12940-022-00860-0. PMID: 35501856).
I would also recommend linking the outcome of these studies to the need for the realisation of new studies exploring the links between glyphosate and human health at environmentally relevant concentrations alone or in combination with other pollutants (see for example Vardakas et al. A Mixture of Endocrine Disruptors and the Pesticide Roundup® Induce Oxidative Stress in Rabbit Liver When Administered under the Long-Term Low-Dose Regimen: Reinforcing the Notion of Real-Life Risk Simulation. Toxics. 2022 Apr 14;10(4):190. doi: 10.3390/toxics10040190. PMID: 35448451).
A number of publications also showed that AMPA but not glyphosate was associated to health outcomes in human populations. The findings that AMPA can reflect other exposure is bringing important insights to show that these studies may be misinterpreting the findings if they conclude that the source of AMPA was glyphosate (e.g. in Lucia et al. Association of Glyphosate Exposure with Blood DNA Methylation in a Cross-Sectional Study of Postmenopausal Women. Environ Health Perspect. 2022 Apr;130(4):47001. doi: 10.1289/EHP10174. Epub 2022 Apr 4. PMID: 35377194).
Reviewer 2 Report
1.Line 83-84 For assessing the determinants of variability of Gly and AMPA concentrations, both variables were dichotomized (value 0 for values below LOQ and value 1 for values above according to the LOQs in the individual studies). The reviewer has difficulty to understand why the variabilities of Gly and AMPA were determined by this way. The authors are suggested to make explanations.
2.Line 110-113 This results in the predicted daily intake of Gly expressed as percentage of the ADI; %ADI =Glyconc x Vol/bw/Fue/ADI, The authors are suggested to explicitly explain what Fue means.
3.Line 180-185 “When regressing AMPA (ln μg/g creatinine) on the Y-axis against Gly (ln μg/g creatinine) on the X-axis, the slope of the linear fit is smaller than 1 (P<0.001). When the regression line is compared with results in children (Figure 3)it can be seen that these are similar. Slopes vary between 0.35 and 0.39 and the intercept between -0.90 and -1.06. This observation indicates the existence of ‘autonomous’ origins of AMPA”. The authors are suggested to explicitly describe the bases to indicate the existence of autonomous origins of AMPA.
4.The authors are suggested to review literature related to the precursors of AMPA in the introduction since the suthors implied other sources of AMPA in the environment in discussion, for example, other sources like amino-polyphosphonates.
Reviewer 3 Report
This paper reports Gly and AMPA mesures in urine sample from 4 national studies in general population. The results are original and interesting, even if ultimately exposure determinant analysis is not contributive.
The manuscript is very pleasant to read. Methodological choices are clearly justified. The controversy about Gly cancerogenicity is well addressed.
However, there is something pretty disturbing about the discussion of results in children, several times, whereas this study is presented openly as dedicated to adults.
I only have a few questions to discuss.
If my understanding is correct, an assumption for calculation of PDI is that ingestion is the only route of exposure ?
Is country a determinant of exposure ? And/or the method of urine sample ?